# Effect of Tempering Temperature after Thermo-Mechanical Control Process on Microstructure Characteristics and Hydrogen-Induced Ductility Loss in High-Vanadium X80 Pipeline Steel

**DOI:** 10.3390/ma13122839

**Published:** 2020-06-24

**Authors:** Longfei Li, Bo Song, Biwen Yang, Lei Wang, Wensen Cheng

**Affiliations:** School of Metallurgical and Ecological Engineering, University of Science and Technology Beijing, Beijing 100083, China; aifei_0105@126.com (L.L.); ybwtx0807@163.com (B.Y.); wanglei_ustb020@163.com (L.W.); cheng765206437@126.com (W.C.)

**Keywords:** tempering temperature, microstructure characteristics, hydrogen-induced ductility loss, high-vanadium X80 pipeline steel

## Abstract

In this study, an optimum tempering temperature after a thermo-mechanical control process (TMCP) was proposed to improve the hydrogen-induced ductility loss of high-vanadium X80 pipeline steel. The results showed that with increasing tempering temperature from 450 to 650 °C, the size and quantity of granular bainite decreased but the spacing of deformed lath ferrite and the fraction of massive ferrite increased. The number of fine vanadium carbides increased as well. However, as the tempering temperature increased to 700 °C, the microstructure of T700 steel completely converted to massive ferrite and the grain size became larger. Additionally, the amount of nanoscale precipitates decreased again, and the mean size of precipitates evidently increased in T700 steel. The steel tempering at 650 °C, containing the most vanadium precipitates with a size less than 20 nm, had the lowest hydrogen diffusion coefficient and the best resistance to hydrogen-induced ductility loss.

## 1. Introduction

Over the past decades, with the continuous exploitation of petroleum and natural gas in remote areas, long distance oil/gas pipeline transportation is facing enormous challenges [1,2]. More and more severe service environment and the self-acidity of oil and gas both put forward more stringent requirements on mechanical properties and corrosion resistance for pipeline steel [3]. After long-term research, hydrogen-induced corrosion failures have been summarized as two main types, hydrogen embrittlement (HE) and hydrogen-induced cracking (HIC) [2,4,5,6]. The mechanisms of hydrogen damage and the methods to improve hydrogen-induced failure have come to light through constant efforts.

A large number of hydrogen atoms penetrating into steel is the main cause of failures. The hydrogen atoms move to the steel surface by reduction of hydrogen ions from external acidic environment and enter steel through physisorption, driven by van der Waals’ forces, and chemisorption successively [7]. For one thing, owing to the small atomic radius, hydrogen atoms diffuse easily in the metal lattice and gather at defects such as large-dimension inclusions. The hydrogen atoms then react with each other to form a large number of hydrogen molecules, which probably increase local internal pressures and cause the formation of hydrogen blister and HIC cracks [8,9,10]. For another, the diffusible hydrogen atoms in steel also result in hydrogen embrittlement. Currently, hydrogen-enhanced decohesion (HEDE) and hydrogen-enhanced localized plasticity (HELP) are two of the most recognized mechanisms to explain hydrogen embrittlement in single-phase steels [9,11]. HEDE theory considers that hydrogen reduces the interatomic cohesive forces between metallic lattices [12,13]. When the maximum normal stress at the crack tip is greater than the interatomic forces, atom pairs are pulled apart and the cracks nucleate, which eventually cause hydrogen embrittlement fracture. A. Nagao et al.’s [13] study revealed the effect of hydrogen atoms on “intergranular” and “quasi-cleavage” fracture surface generation along prior austenite grains and lath boundaries in lath martensitic steels. However, HELP theory holds that the presence of hydrogen decreases the activation energy for dislocation motion and the equilibrium separation distance between dislocations in the pile-ups [9,14,15,16], which promotes the emission of dislocations and enhances localized plasticity [17,18,19,20]. Besides, HELP mechanism is also invoked to understand the reduction of material yield stress and some softening phenomenon in the presence of hydrogen [21]. The localization of deformation and the reduction of cross-slip are in agreement with the shielding model of the HELP mechanism, as I.M. Robertson et al. [22,23] reported.

Nonetheless, many researchers have conducted different investigations on controlling the quantity and distribution of diffusible hydrogen in steel so as to improve the resistance to HIC and HE failures. Microstructures could play a significant role in hydrogen diffusion and susceptibility to HE. It was confirmed that fine-grained massive ferrite and acicular ferrite with lower hydrogen diffusivity had better performance on impeding HE behavior than coarse-grained granular bainite and bainite ferrite [24]. Generally, hydrogen atoms were likely to be trapped at the interfaces between the retained austenite and martensite. Thus, the M-A constituent was more likely to cause crack initiation due to its high hardness and brittleness [25,26,27]. In addition, other microstructural parameters like coincidence site lattice (CSL) boundaries and the recrystallization fraction are also considered as important factors affecting HIC crack propagation. M.A. Mohtadi-Bonab et al. [28] found that Σ3 type boundary increased HIC susceptibility while the large amounts of recrystallization fraction with no stored energy brought about high HIC resistance. Equally, crystallographic texture is deemed another main factor determining HIC crack propagation. M.A. Arafin et al. [29] verified through their studies that the grains with <100>//ND orientations made the pipeline steel more susceptible to HIC. Furthermore, it is notable that fine nanoscale precipitates can be used as hydrogen traps to effectively diminish the amount of diffusible hydrogen atoms. A. Nagao et al. [30] concluded that the nanosized (Ti, Mo) C precipitates contributed to make less hydrogen atoms reassign to the grain boundaries by lattice diffusion and dislocations. The highly dispersed nanoscale NbC precipitates, as irreversible traps, hindered hydrogen diffusion and aggregation, which improved the resistance to HIC crack initiation and propagation, as reported by X.G. Li et al. [31,32]. T. Depover et al. [14,33,34,35,36,37,38] summed up the effects of five different types of carbides, i.e., Ti, V, Mo, Cr and W based carbides, on HE susceptibility. They found that the tempering-induced TiC and V_4_C_3_ trapped a large amount of hydrogen, while only a limited amount of hydrogen was trapped by Mo_2_C and Cr_23_C_6_ and no hydrogen was trapped by W_2_C precipitates. K. Kawakami et al. observed that vanadium carbides captured most of the deuterium (D) atoms by a three-dimensional atom probe (3DAP) [39] and deduced that the carbon vacancy of V_4_C_3_ had a large trapping energy of hydrogen through first-principles calculations [40]. At present, in order to obtain appropriate microstructure, thermomechanical controlled processing (TMCP) and tempering is a common procedure in pipeline steel production. What is more, tempering process parameters, such as tempering temperature and holding time, have important effects on the microstructure types, grain size and precipitation behavior of steel. However, few research works have been conducted to understand the effect of tempering process on the resistance to hydrogen-induced failures in X80 pipeline steel that has been widely applied to oil and gas transportation.

Therefore, the present work was aimed to understand the influence of tempering temperature after TMCP on the characteristics of microstructures, grain boundaries and orientation, nanoscale precipitates and further on hydrogen-induced ductility loss in high-vanadium X80 pipeline steel. Focus was put on the relationship between microstructural characteristics and hydrogen diffusion in steel under different tempering temperatures.

## 2. Materials and Method

The chemical composition of the high-vanadium X80 pipeline steel used in this study is demonstrated in Table 1. The steel was melted in a 50 kg vacuum induction furnace, and then the ingot was forged into a billet of size 240 × 100 × 45 mm^3^. Subsequently, after two-pass controlled rolling, the plate thickness reduced to 12 mm. The six experimental steels were tempered at different temperatures (450 °C, 500 °C, 550 °C, 600 °C, 650 °C and 700 °C) for 30 min, numbered as T450, T500, T550, T600, T650 and T700 respectively. The specific hot working processes are shown in Figure 1.

The microstructures and grain orientation on the plane perpendicular to the normal direction (ND) were investigated using a field emission scanning electron microscopy (FE-SEM) and a Zeiss Gemini SEM-500 equipped with an Oxford Instruments Nordlys nano electron backscatter diffraction (EBSD) detector. The metallographic samples were ground, polished, etched with 4% nital solution. The detection area for EBSD analysis was about 160 × 120 μm^2^ with a step size of 0.2 μm at 20 kV. Precipitation particles were analyzed by JEM high resolution transmission electron microscope (HRTEM). For this purpose, the carbon replicas for six steels were prepared. Thin carbon films were sprayed on the surfaces of samples, which were firstly ground, polished and etched with a 4% nital solution. Then, they were cut into small blocks with the sizes of 2 × 2 mm^2^ and extracted on specific copper nets by immersing in a 6% nital solution. The size and distribution of precipitates were analyzed by Image J software. The same statistical areas (5.74 μm^2^) of the six tested steels taken randomly were measured to ensure the accuracy of statistics results.

The hydrogen permeation tests were conducted in a Devanathan–Stachurski electrochemical cell. The steel membranes with sizes of 20 × 20 × 1 (ND) mm^3^ were polished and a uniform nickel coating was electroplated on the detection side to prevent the formation of oxide layer. The steel membrane was placed between the two electrolysis cells and a circular area of 1.76 cm^2^ was exposed to the two cells. The solution of the detection cell was a deaerated 0.1 M sodium hydroxide (NaOH) while the solution of the charging cell included 0.5 vol.% glacial acetic acid, 3.5 wt % sodium chloride and 1 g/L sodium sulfide (Na_2_S·9H_2_O). The specific experiment device schematic diagram is depicted in Figure 2 [41].

The physical parameters were calculated by the following Formulas (1)–(3) [42,43,44]. The hydrogen flux *J*_∞_ (mol·cm^−1^·s^−1^) through the steel membrane and the effective diffusion coefficient *D*_eff_ were respectively determined by steady-state current, *I*_∞_, and the breakthrough time, *t_b_*, whereas the apparent hydrogen concentration *c*_app_ was obtained by the values of *J*_∞_ and *D*_eff_. In addition, the hydrogen trap density *N*_T_ can be evaluated by Formula (4) [45].
(1)$$J_{\infty }L=\frac{I_{\infty }\cdot L}{F\cdot A}$$
(2)$$D_{\mathrm{eff}}=\frac{L^{2}}{15.3t_{b}}$$
(3)$$c_{\mathrm{app}}=\frac{J_{\infty }L}{D_{\mathrm{eff}}}$$
(4)$$N_{T}=\frac{c_{\mathrm{app}}}{3}\times \left(\frac{D_{L}}{D_{\mathrm{app}}}-1\right)$$
where *L* is the membrane thickness, *F* is the Faraday constant, *A* is the work area exposed, *N*_T_ is the number of hydrogen trapping sites per unit volume and D_L_ is the lattice diffusion coefficient. In α-Fe matrix, D_L_ = 1.28 × 10^−4^ cm^2^·s^−1^.

Additionally, based on the tensile test results at room temperature with a tensile rate of 2 mm/min in hydrogen pre-charging and in air conditions, the hydrogen embrittlement susceptibility index I_HE_ was expressed by Formula (5) [46]. The sizes of tensile specimens taken from the steel plates parallel to normal plane at 1/4 of thickness are shown in Figure 3. Hydrogen was charged into specimens using a 250 mg/L arsenic oxide (As_2_O_3_) in a 0.5 M H_2_SO_4_ acid corrosive solution at a current density of 10 mA·cm^−2^ for 1 h to avoid creating blisters or other defects.
(5)$$I_{\mathrm{HE}}=\left(1-\frac{\delta_{H}}{\delta_{0}}\right)\times 100\%$$
where *δ*_0_ is the elongation in air, and *δ*_H_ is the elongation in the hydrogen pre-charging condition.

## 3. Results and Discussion

### 3.1. Evolution of Microstructures and Crystallization Orientation

Figure 4 shows representative microstructures of the experimental steels with different tempering temperature after hot-rolling. Obviously, the microstructure of T450 steel was composed of massive ferrite, deformed ferrite and a larger proportion of granular bainite, as depicted in Figure 4a. With increasing tempering temperature, the size and quantity of granular bainite gradually decreased, while the spacing of deformed lath ferrite and the area of massive ferrite increased. In particular, when tempering temperature increased to 700 °C, granular bainite and deformed lath ferrite in steel almost disappeared and the size of massive ferrite sharply increased. In the meantime, there were some coarsened nanoscale carbides nearby the grain boundary. It was because the increasing tempering temperature after hot-rolling promoted the thermal motion of carbon atoms in bainite and these interstitial atoms were easier to diffuse into steel matrix. Additionally, at higher tempering temperatures, the diffused carbon could bind to microalloying elements and further form carbide precipitates. These fine precipitates could not only enhance strength by impeding dislocation movements, but also reduce the amount of diffusible hydrogen atoms as effective hydrogen traps.

The inverse pole figure (IPF) maps on the plane perpendicular to ND of the six as-received X80 high-vanadium pipeline steels are respectively depicted in Figure 5a–f. High angle grain boundaries (HAGBs) with a misorientation angle of 15° < θ < 62.8° were indicated by thick-black lines, while thin-black lines corresponded to low angle grain boundaries (LAGBs) with a misorientation angle of 3° < θ < 15°. The fraction of the high angle grain boundary and the average grain size were calculated in the corresponding area as shown in Figure 6. The fraction of HAGBs slightly reduced with increasing tempering temperature from 450 to 650 °C. However, the fraction of HAGBs in T700 steel increased significantly, even more than 4 times of that in T650 steel. On one hand, when tempering temperature reached 700 °C, the fraction of deformed ferrite evidently decreased. The deepened recovery degree led to the dislocation slipping and climbing and further the annihilation of dislocations each other, which brought a decrease in the fraction of LAGBs in ferrite microstructure. On the other hand, the nanoscale precipitates in T700 steel coarsened obviously, which weakened their role in hindering dislocation movement and pinning grain boundaries. Furthermore, it caused great increase in crystal grain size.

Additionally, previous studies [28] concluded that the number of atoms arranged on the special CSL grain boundaries between Σ5 and Σ13b were larger, which could decrease the vacancy density and further weaken the possibility of hydrogen atom segregation at the grain boundary. As a consequence, this type of grain boundary would play a significant role in intergranular HIC resistance in pipeline steel. Hence, the CSL boundary distribution of the six experimental steels was analyzed by EBSD, as shown in Figure 7. It was notable that the frequency of the special CSL boundaries between Σ5 and Σ13b in steel after tempering at 700 °C was higher than the other steels. After tempering at higher temperature, the dislocations near the low angle grain boundaries recovered, which promoted the formation of the CSL grain boundaries with lower energy. Besides, it is well-accepted that the boundaries around the grain with <111>//ND orientations make highly resistant to HIC while the boundaries around the grain with <100>//ND orientations are easier to provide the paths for HIC crack propagation [28,29]. Here, Figure 8a–f demonstrates the EBSD analysis results for the special <111>//ND orientation of the experimental steels. It can be seen that as the tempering temperature after hot-rolling increased from 450 to 550 °C, the component of <111>//ND orientation increased from 5.52% to 15.70%. However, the values fluctuated in a small range when the tempering temperature continued to rise. This kind of special grain was distributed evenly in all steels.

### 3.2. Characteristics of Nanoscale Precipitates

Figure 9a–f respectively shows the typical carbide morphology of different experimental steels. Additionally, the size distributions of nanoscale precipitates were analyzed through statistics for TEM images, as shown in Figure 10. With increasing tempering temperature after hot-rolling, the number of fine precipitates increased first and then decreased, and T650 steel had the largest number of fine precipitates. Here, the effective replica thickness is assumed to be equal to the mean diameter of particles, *D*_mean_. The volume fractions of the nanoscale particles, V_f_, were calculated using modified McCall-Boyd Formula (6) as follows [47,48]:(6)$$V_{f}=\left(\frac{1.4\pi }{6}\right)\times \left(\frac{ND_{\mathrm{mean}}^{2}}{A}\right)$$
where *N* is the number of precipitates in the statistical area and A is the statistical area.

Based on the statistical results from TEM images, the number in the statistical area, the mean diameters and the volume fraction of precipitates are given in Table 2. The V_f_ values were mainly affected by the number of precipitates and it was the highest in T650 steel. In addition, with increasing tempering temperature, the mean diameter of precipitates gradually increased. Especially, the size of precipitates increased significantly as the tempering temperature after hot-rolling rose to 700 °C, which corresponded to the precipitate size distribution results in Figure 10. The proportion of precipitates with a size larger than 20 nm in T700 steel was remarkably higher than other experimental steels. It was because the Oswald ripening of fine carbides was more likely to occur in the steel tempered at higher temperature. The Ostwald ripening of common second phase particles in steel mostly obeyed the 1/3 power law and it depended on the diffusion of solute atoms in matrix [49,50]. The ripening rate of the specific carbide, *m*_C_ can be calculated by Formula (7):(7)$$m_{C}={\left(\frac{8\sigma V_{P}^{2}Dc_{0}}{9V_{B}c_{P}RT}\right)}^{1/3}$$
where *σ* is the interface energy between the carbide and matrix, *V*_P_ and *V*_B_ are respectively the molar volume of carbide and microalloyed elements, *c*_0_ and *c*_P_ are respectively the concentrations of microalloyed elements in matrix and in the carbide, D is the element diffusion coefficient and R is the gas constant. According to the results of previous studies [51,52], the values of σ for the carbides containing V, Nb and Ti can be obtained by Equations (8)–(10) and the diffusion coefficient of these three elements in α-Fe can be calculated by Equations (11)–(13). After substituting the Equations (8)–(13) and the physical parameters of microalloying elements and corresponding carbides in Table 3 into the Equation (7), the changes in the ripening rates with temperature could be derived and depicted in Figure 11. With increasing temperature, the ripening rate of carbides exponentially increased. What is more, vanadium carbides were easier to grow up than the other two kinds of carbides. It explained why the particles in T700 steel with high vanadium content had the largest mean diameter.
(8)$$\sigma_{\mathrm{VC}-\alpha }=0.8055-0.2677\times 10^{-3}T$$
(9)$$\sigma_{\mathrm{NbC}-\alpha }=1.2537-0.4166\times 10^{-3}T$$
(10)$$\sigma_{\mathrm{TiC}-\alpha }=1.0687-0.3552\times 10^{-3}T$$
(11)$$D_{V-\alpha }=3.92\exp\left(-\frac{241000}{RT}\right)$$
(12)$$D_{\mathrm{Nb}-\alpha }=50.2\exp\left(-\frac{252000}{RT}\right)$$
(13)$$D_{\mathrm{Ti}-\alpha }=3.15\exp\left(-\frac{248000}{RT}\right)$$

Figure 12a–c reveals the morphology and types of the nanoscale precipitates with different sizes and positions in T650 steel using the HRTEM analysis. After measuring lattice spacing and comparing it with the standard crystallographic parameters, it can be seen that the rod-like particle about 10 nm precipitated along the grain boundaries, was vanadium carbide (VC), while the round one with size less than 10 nm were confirmed to be another type of vanadium carbide (V_8_C_7_) that nucleated in grains. That was, the particles near the grain boundaries were generally larger than the intragranular ones.

### 3.3. Hydrogen Diffusion Behavior

According to the hydrogen permeation tests, the variation curves of hydrogen oxidation current vs. time for the six different samples are shown in Figure 13. The parameters calculated by Formulas (1)–(4) for evaluating hydrogen diffusion behavior are listed in Table 4. With the increase of tempering temperature after hot-rolling from 450 to 650 °C, the effective hydrogen coefficient decreased. However, as the tempering temperature continued to rise to 700 °C, the effective hydrogen coefficient increased again. The change trend of the hydrogen flux J_∞_ was similar to the effective hydrogen coefficient. Besides, the apparent hydrogen concentration among the four experimental steels increased first and then decreased with increasing tempering temperature and T650 steel had the highest hydrogen concentration. Similarly, T650 steel also had the maximum hydrogen trap density, 7.06 × 10^19^ cm^−3^. Based on the precipitate analysis results, more and more nanoscale vanadium carbide particles precipitated in matrix with the increase of tempering temperature from 450 to 650 °C. The number of the carbides in T700 steel decreased and their size obviously increased. These fine precipitates can act as hydrogen traps to fix the diffusible hydrogen atoms in steel, which decreased the effective hydrogen coefficient and increased the apparent hydrogen concentration. However, as Depover et al. reported [33], carbides with a size above 20 nm would not play a determinant role in terms of trapping. This was the main reason why hydrogen diffused more easily in T700 steel than T650 steel. What is more, some researchers found that grain boundaries could capture diffusible hydrogen atoms in steel acted as reversible hydrogen traps under no external stress condition [28]. In T700 steel, excess tempering temperature resulted in grains coarsening and the reduction of grain boundary number per unit area. They both weakened the hydrogen trap efficiency of grain boundary in steel.

### 3.4. Hydrogen Induced Mechanical Degradation

Figure 14 shows the stress–strain curves of the experimental steels in hydrogen pre-charging and in air conditions. The hydrogen embrittlement susceptibility indexes calculated by Formula (5) and the steel elongations are listed in Table 5. With the increase of tempering temperature after hot-rolling, the elongations of the specimens without hydrogen pre-charging decreased. Especially, when the tempering temperatures rose from 650 to 700 °C, the elongation decreased drastically. However, for the tensile specimens after hydrogen pre-charging with tempering temperature in the range of 450–650 °C, the elongations were basically at the same level, whereas the elongation of T700 steel after hydrogen pre-charging was much lower than others. Therefore, the hydrogen embrittlement sensitivity of T650 steel was the minimum, while T700 steel had the worst resistance to hydrogen-induced ductility loss.

Figure 15 shows the tensile fracture topographies of the six different experimental steels after hydrogen pre-charging. It can be seen that all the fractures were plastic and contained a large number of dimples. The fractures of the specimens tempered at the temperature range of 450–650 °C had larger and deeper dimples, and most of them were equiaxed. However, the number and depth of dimples at the fracture of T700 steel evidently decreased, and there were large and smooth areas between dimples. It indicated that the energy was released instantaneously once the fracture occurred, and T700 steel after hydrogen pre-charging had the worst plasticity.

At present, hydrogen-enhanced decohesion (HEDE) and hydrogen-enhanced localized plasticity (HELP) mechanisms were two main methods to explain the hydrogen-induced ductility loss. The HEDE mechanism considered that hydrogen atoms in the lattice weakened the bond strength, while the HELP mechanism thought that the attachment of hydrogen at dislocations led to their emission, motion and enhanced localized plasticity. It could be concluded that between these two theories there was a common feature that diffusible hydrogen atoms in matrix was the main cause of failures. In view of the results from the hydrogen permeation tests, although the apparent hydrogen concentration of T650 steel was the highest, most hydrogen atoms were fixed by fine vanadium carbides. On the contrary, the number of diffusible hydrogen atoms in steel was the lowest owing to its lowest hydrogen diffusion coefficient. This effectively weakened the kinetic conditions of hydrogen-induced ductility loss. It was worth noting that the ability of nanoscale vanadium carbides acted as irreversible hydrogen traps to fix hydrogen atoms was much better than that of grain boundaries and dislocations with low hydrogen binding energy [14]. Under a certain stress, grain boundaries and dislocations as traps would desorb the hydrogen atoms that might be diffusible again and further cause hydrogen embrittlement failure. Additionally, the hydrogen embrittlement sensitivity index was also affected by the inherent plasticity that was decided by grain size and precipitate characteristics. The steels containing small grains were not prone to brittle fracture because they had low deformation inhomogeneity and small stress concentration areas. The tensile specimens can endure marked plastic straining before fracture. Beyond that, in T700 steel, the coarsened non-deformable second-phase particles reduced the uniform plasticity by impeding dislocation movement. It was well explained that T700 steel with much larger grain size than other experimental steels had the lowest elongation. Moreover, a large amount of carbides with sizes above 20 nm in T700 had weaker hydrogen capture capability, which reduced the role on improving the resistance of steel to hydrogen-induced ductility loss. Although, based on the above analysis, T700 steel had the highest fraction of HAGBs and the special CSL boundaries between Σ5 and Σ13b both with good resistance to crack propagation, it cannot compensate for the deterioration of the hydrogen-induced ductility loss caused by larger grain size and less effective hydrogen traps. Therefore, the hydrogen embrittlement sensitivity index of T700 steel increased significantly. Nevertheless, T650 steel containing the largest number of nanoscale vanadium carbides had the optimum resistance to hydrogen-induced ductility loss.

## 4. Conclusions

In this study, the effect of tempering temperature after TMCP on the precipitates, microstructures and crystallographic orientation was analyzed. Furthermore, the roles of these characteristics on hydrogen diffusion behavior and hydrogen-induced ductility loss were investigated systematically. The conclusions were summed up as follows:With increasing tempering temperature from 450 to 650 °C, the size and quantity of granular bainite decreased gradually, while the spacing of deformed lath ferrite and the area of massive ferrite increased. The microstructure of T700 steel was almost completely composed of massive ferrite and its grain size increased obviously.With the increase of tempering temperature, the number of fine vanadium carbides increased first and then decreased again. T650 steel tempered at 650 °C had the largest amount of nanoscale precipitates, while the mean size of precipitates increased in T700 steel.T650 steel containing more dispersed nanoscale vanadium carbides with size less than 20 nm had the lowest hydrogen diffusion coefficient and the optimum resistance to hydrogen-induced ductility loss.

## Figures and Tables

**Figure 1 materials-13-02839-f001:**
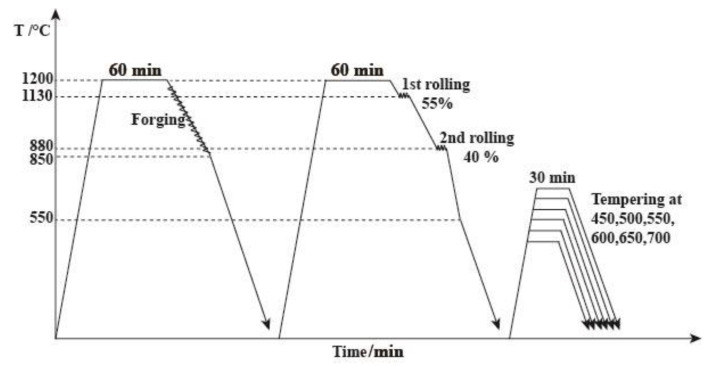
Schematic graph of thermomechanical controlled processing (TMCP) and tempering processes.

**Figure 2 materials-13-02839-f002:**
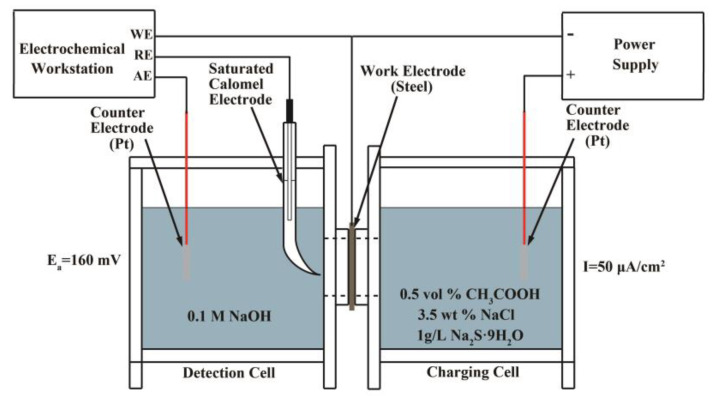
The Devanathan-Stachurski setup used for hydrogen permeation test.

**Figure 3 materials-13-02839-f003:**
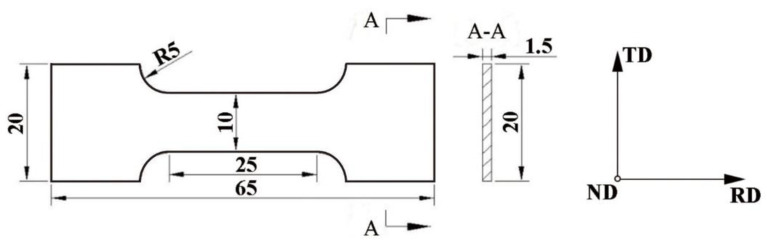
Schematic diagram of the tensile specimen used in the tension test with pre-charging and in air conditions.

**Figure 4 materials-13-02839-f004:**
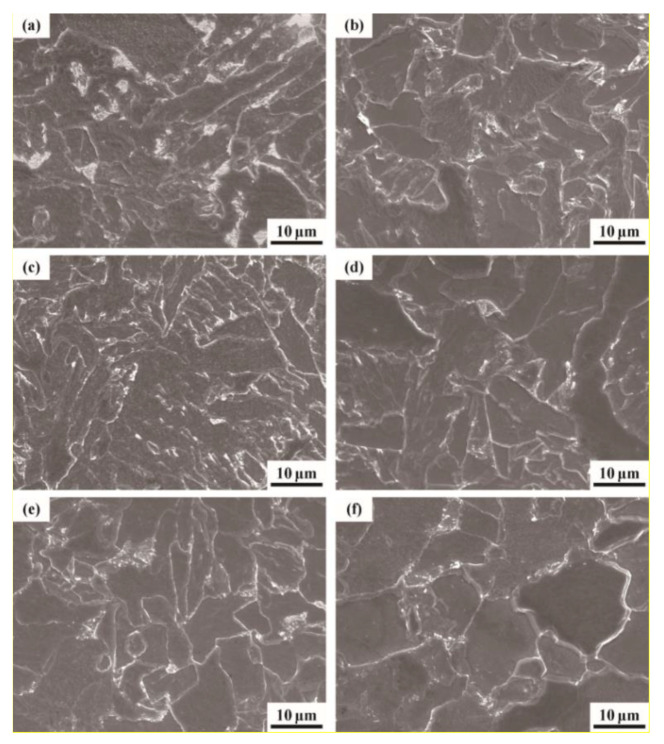
SEM images of the experimental steel microstructures: (**a**) T450; (**b**) T500; (**c**) T550; (**d**) T600; (**e**) T650 and (**f**) T700.

**Figure 5 materials-13-02839-f005:**
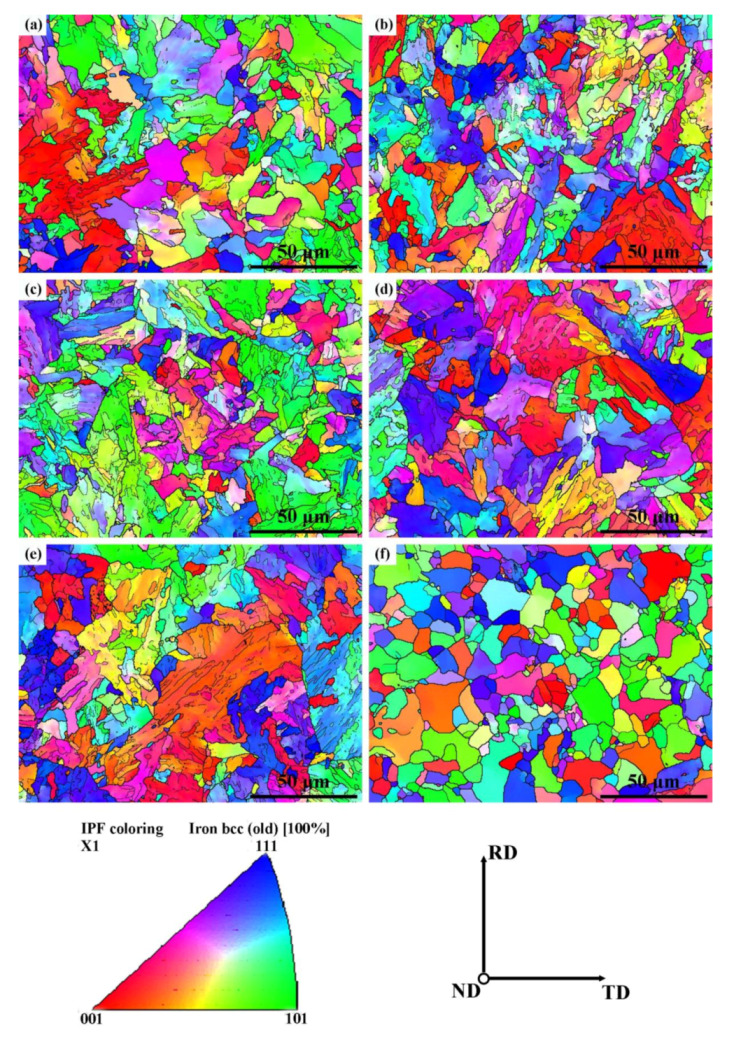
Inverse pole figure (IPF) images of the six experimental steels: (**a**) T450; (**b**) T500; (**c**) T550; (**d**) T600; (**e**) T650 and (**f**) T700.

**Figure 6 materials-13-02839-f006:**
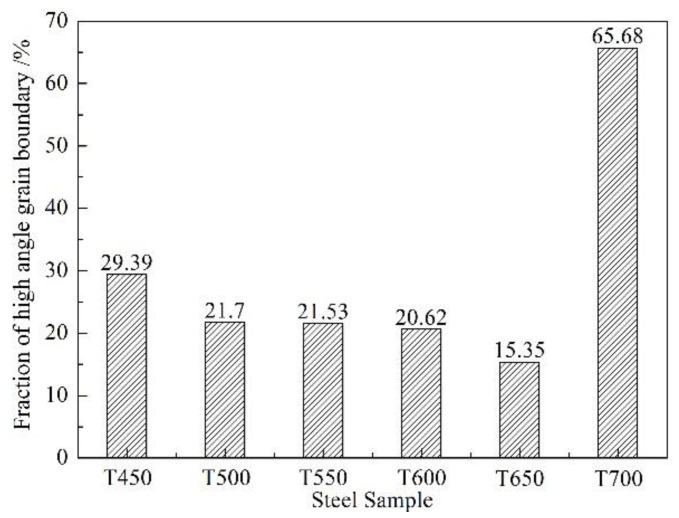
High angle grain boundary (HAGB) fractions of the experimental steels.

**Figure 7 materials-13-02839-f007:**
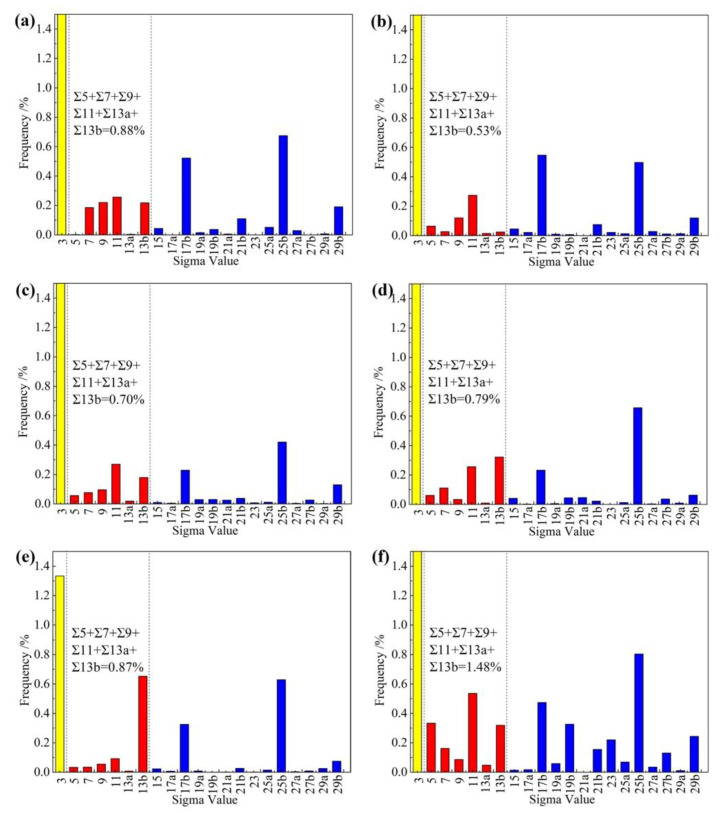
Coincidence site lattice (CSL) boundary distributions in the experimental steels: (**a**) T450; (**b**) T500; (**c**) T550; (**d**) T600; (**e**) T650 and (**f**) T700.

**Figure 8 materials-13-02839-f008:**
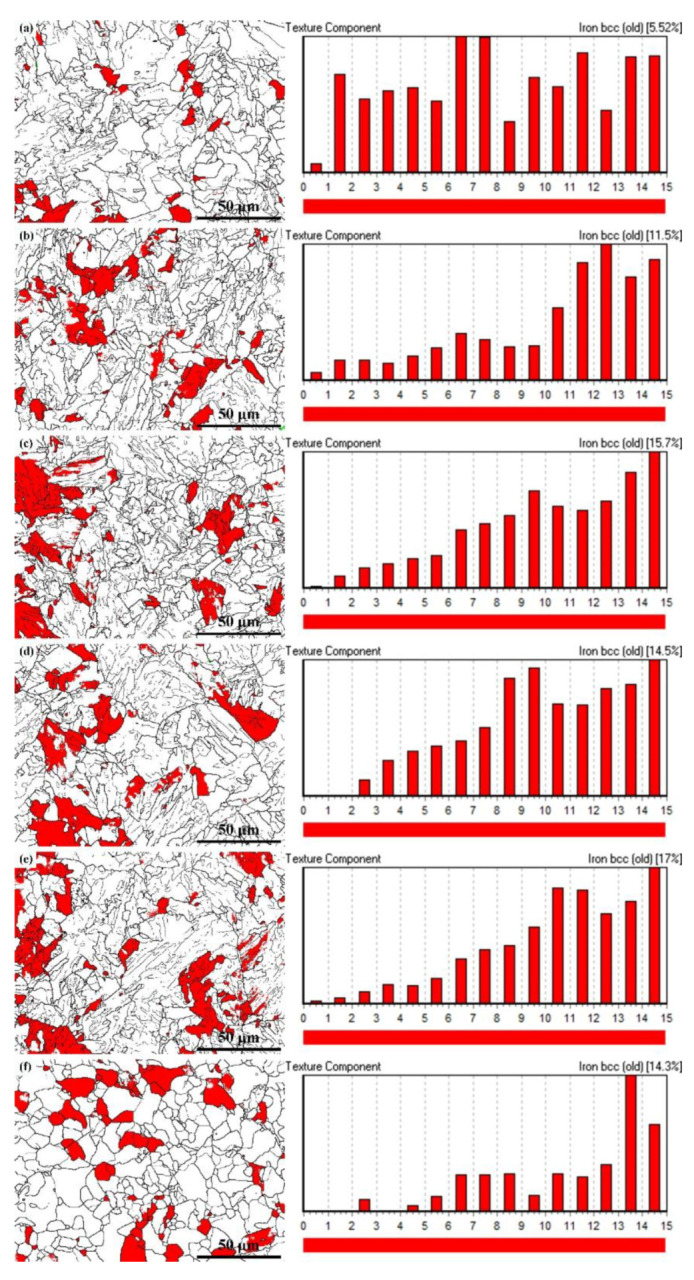
Electron backscatter diffraction (EBSD) analysis results for the special <111>//ND orientation marked with red: (**a**) T450; (**b**) T500; (**c**) T550; (**d**) T600; (**e**) T650 and (**f**) T700.

**Figure 9 materials-13-02839-f009:**
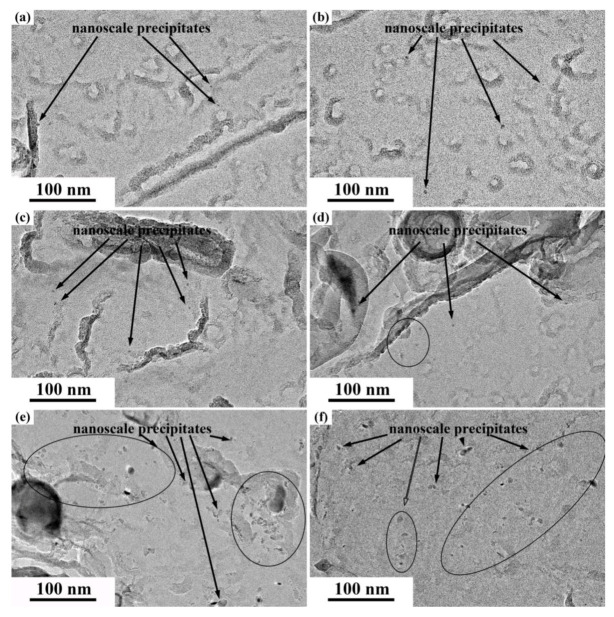
TEM bright field images of precipitates morphology on carbon replicas films: (**a**) T450; (**b**) T500; (**c**) T550; (**d**) T600; (**e**) T650 and (**f**) T700.

**Figure 10 materials-13-02839-f010:**
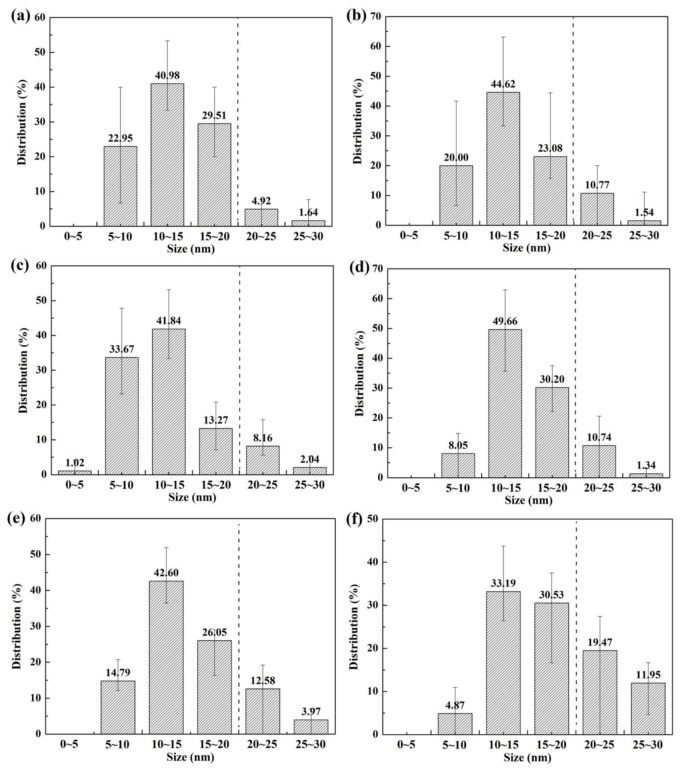
The size distribution of nanoscale precipitates for the experimental steels: (**a**) T450; (**b**) T500; (**c**) T550; (**d**) T600; (**e**) T650 and (**f**) T700.

**Figure 11 materials-13-02839-f011:**
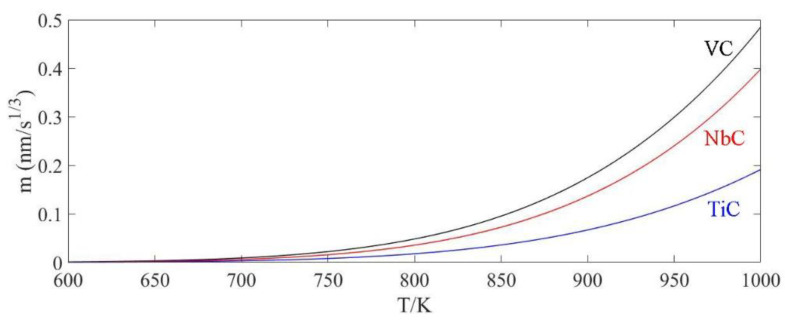
Relationship between the ripening rate of precipitate and temperature.

**Figure 12 materials-13-02839-f012:**
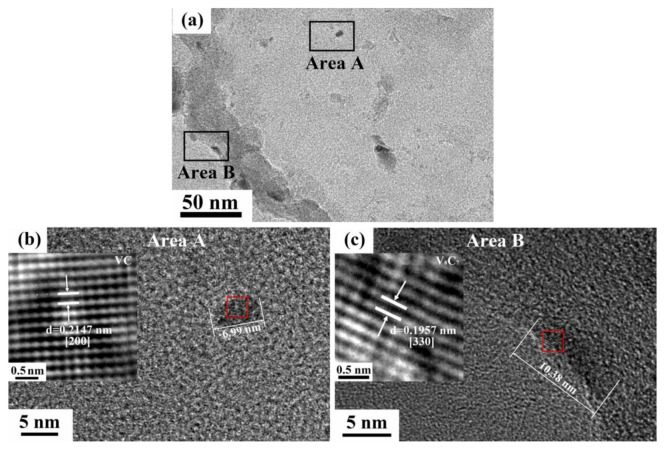
HRTEM analysis carbide precipitates in T650 steel: (**a**) morphology of precipitates at low magnification and (**b**,**c**) high resolution images of nanoscale precipitates corresponding to area A and area B in (**a**).

**Figure 13 materials-13-02839-f013:**
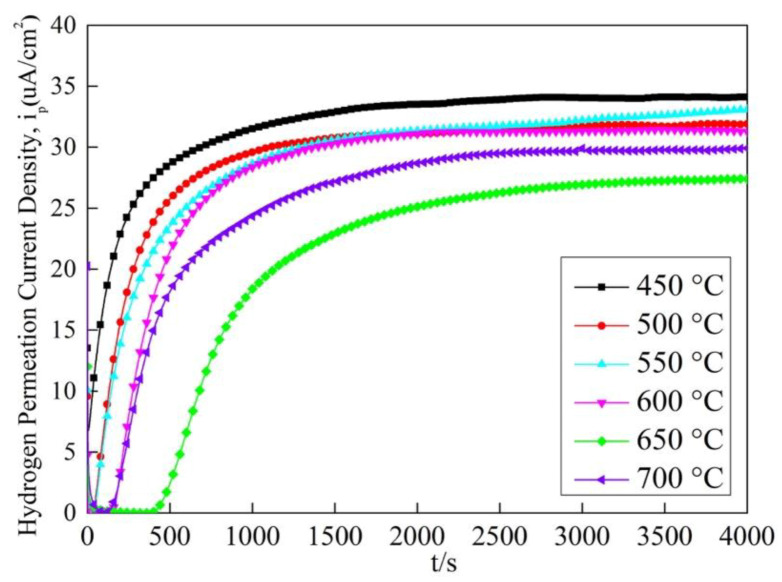
Hydrogen permeation curves for the experimental steel.

**Figure 14 materials-13-02839-f014:**
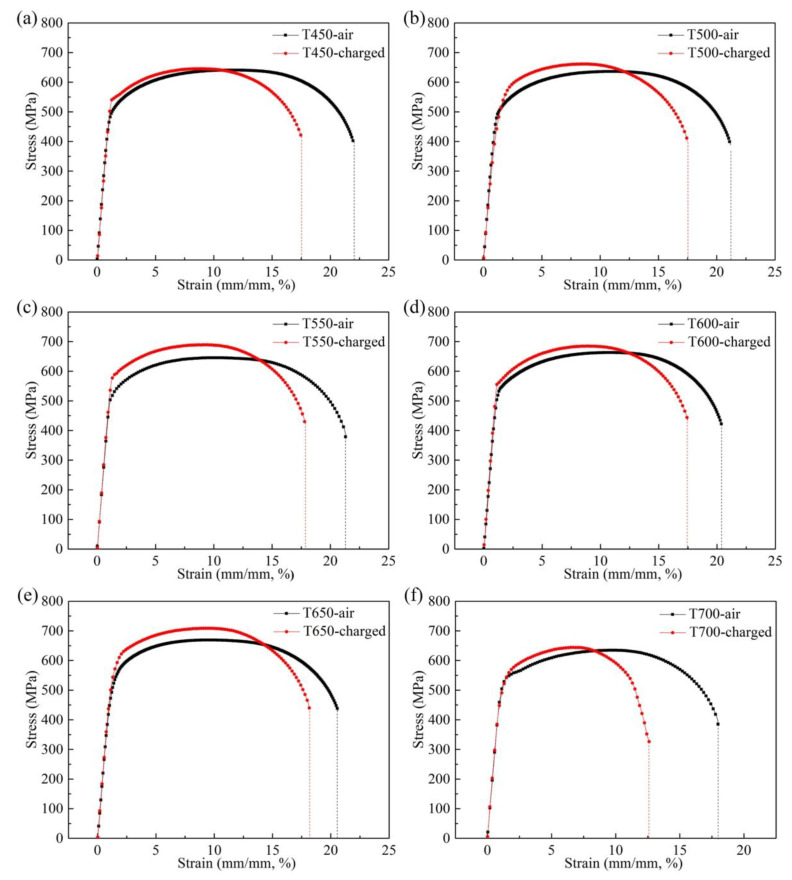
Stress–strain curves of tensile specimens in hydrogen pre-charging and in air conditions: (**a**) T450; (**b**) T500; (**c**) T550; (**d**) T600; (**e**) T650 and (**f**) T700.

**Figure 15 materials-13-02839-f015:**
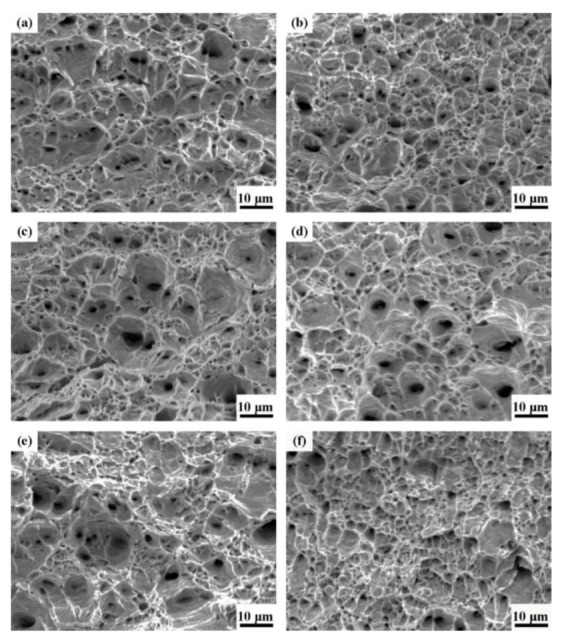
SEM images of tensile fractographies in hydrogen pre-charging condition: (**a**) T450; (**b**) T500; (**c**) T550; (**d**) T600; (**e**) T650 and (**f**) T700.

**Table 1 materials-13-02839-t001:** The chemical composition of the material/wt %.

C	Si	Mn	P	S	Ni	Cr	Mo	V	Nb	Ti	Al	O	N	Fe
0.034	0.15	1.80	0.0018	0.004	0.22	0.25	0.11	0.12	0.021	0.005	0.03	0.005	0.0033	Bal

**Table 2 materials-13-02839-t002:** The mean diameter and volume fraction of precipitates for the experimental steels.

Statistical Parameters	Sample Number					
	T450	T500	T550	T600	T650	T700
*N*	61	65	98	149	453	226
D_mean_/(nm)	13.76 ± 4.21	13.89 ± 4.33	14.08 ± 5.02	14.83 ± 4.00	14.96 ± 4.91	17.53 ± 5.39
V_f_/(‰)	2.60	2.82	4.37	7.37	22.79	15.61

**Table 3 materials-13-02839-t003:** Physical parameters of microalloying elements and corresponding carbides [53].

Element and Phases	V_B_/(m^3^·mol^−1^)	V_P_/(m^3^·mol^−1^)	c_0_
V	0.837 × 10^−5^	——	——
Nb	1.085 × 10^−5^	——	——
Ti	1.014 × 10^−5^	——	——
VC	——	1.091 × 10^−5^	1.318 × 10^−3^
NbC	——	1.381 × 10^−5^	1.167 × 10^−4^
TiC	——	1.215 × 10^−5^	1.806 × 10^−4^

**Table 4 materials-13-02839-t004:** Physical parameters of the permeation transients for the tested steel.

Sample Number	J_∞_L/(mol·cm^−1^·s^−1^)	D_eff_/(cm^2^·s^−1^)	C_app_/(mol·cm^−3^)	N_T_/(cm^−3^)
T450	5.17 × 10^−11^	2.13 × 10^−5^	2.42 × 10^−6^	2.42 × 10^18^
T500	4.83 × 10^−11^	1.25 × 10^−5^	3.84 × 10^−6^	7.10 × 10^18^
T550	5.05 × 10^−11^	9.82 × 10^−6^	5.14 × 10^−6^	1.24 × 10^19^
T600	4.78 × 10^−11^	8.14 × 10^−6^	5.88 × 10^−6^	1.74 × 10^19^
T650	4.15 × 10^−11^	3.82 × 10^−6^	1.08 × 10^−5^	7.06 × 10^19^
T700	4.52 × 10^−11^	6.84 × 10^−6^	6.61 × 10^−6^	2.35 × 10^19^

**Table 5 materials-13-02839-t005:** Summary of hydrogen embrittlement index and elongations of tensile specimens in hydrogen pre-charging and in air conditions.

Sample Number	δ_0_/%	δ_H_/%	I_HE_/%
T450	21.98	17.50	20.38
T500	21.20	17.47	18.06
T550	21.32	17.80	16.51
T600	20.39	17.46	14.37
T650	20.60	18.17	11.80
T700	17.99	12.62	29.85

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
