# Peer review of "Effect of Tempering Temperature after Thermo-Mechanical Control Process on Microstructure Characteristics and Hydrogen-Induced Ductility Loss in High-Vanadium X80 Pipeline Steel"

_materials, 2020, doi:10.3390/ma13122839_

Round 1

Reviewer 1 Report

The present paper reports the investigation of the effect of annealing temperature on microstructural features and hydrogen embrittlement in x80 pipeline steel. The authors used EBSD to characterize the matrix grain boundaries and TEM to identify and quantify the precipitate populations. The authors also measured hydrogen permeation and mechanical properties after hydrogen charging. They concluded that the annealing temperature affected the grain boundary angle distribution and the carbide size and number. They tied those microstructural features to the observed diffusion and mechanical properties to conclude on the optimal annealing temperature for x80 steel.

This paper presents an interesting and novel contribution to the metallurgy of x80 steel and hydrogen trapping in steel. However, the paper in its present state shows a number of shortcomings that should be adderessed before the paper can be considered for publication, as listed below:

General comments:

-The Materials and Method section lacks details regarding the size of the workpiece before and after forging. Specifying the material thickness after final rolling seems preferable too.

-It also lacks the details regarding replica preparation. The entire procedure should be described (initial surface preparation, one or two step method, etching reagent, carbon deposition parameters, cleaning procedure etc). These details are all the more important that micrographs from those replicas are used for quantitative analysis.

-This section should also include all details pertaining to the quantitative analysis of precipitate populations. How was it ensured that the images that were used for the quantitative analysis were recorded at truely random positions ? What are the image processing steps that were used to quantify the different precipitate population parameters? Statistical area and pixel size should also be specified.

-A significant portion of section 3.2 is taken up by a discussion of the precipitate coarsening behavior. However, the authors do not present any evidence that the precipitate populations reached the coarsening stage after the annealing treatments studied here. This point should be clarified.

-It is difficult to follow the scientific message of the paper. In particular, it is claimed by the author as the final point of the conclusion that all microstructural features align with T650 presenting the highest resistance to hydrogen embrittlement. Given the present structure of the paper where results and discussion are lumped in a single section, this conclusion is not brought out clearly. This owed to the large amount of results that the authors present, which is a positive aspect but makes such a simple structure inappropriate. It seems preferable for all results to be laid out first followed by their discussion in a distinct section.

Specific points:

l.241: The reference for the McCall-Boyd formula does not seem to be appropriate as it is simply another paper from the authors where they used the same formula without any justification or reference. It appears that they modified the equation compared to Seher and Maniar, Metallography 5, 409-414 (1972) for instance. The details and hypotheses of this modification should be explicited.

Figure 12: Direct measurement of lattice spacings on HRTEM pictures is not the most reliable way to identify crystals. An analysis of the Fourier transform or better yet, a diffraction pattern would be a lot more convincing. In any case, references for the crystal structure and lattice parameters of the various precipitates should be provided.

l.102-112: Too much experimental detail is given here, this part should focus on scientific question(s) instead.

Table 1: It should be specified that the mass fractions are given as percentages.

Table 2: The meaning of N is specified neither in the caption nor directly in the table. Even if it is specified in the text, it is preferable to include it directly in either as well.

Figure 9: The micrographs in Figure 9 do not serve the purpose for which they are referred to (l. 235). First, the type of contrast should be indicated in the caption (bright or dark field). Second, it is quite difficult to distinguish the carbides that were considered, possibly due to too low a magnification. Markings and annotations could also be used to highlight the carbides.

Figure 13: It is difficult to distinguish the different curve on the current version of the figure, notably for those that are in black or dark blue. Use of fewer but larger symbols could help.

l.285-287: The authors suggest that the large particles visible in figure 12a is a chromium-rich carbide, based on the EDS spectrum in 12b. This conclusion is suprising given the much higher K line intensities seen for Fe and Mn than for Cr. This would rather suggest that this particle, even though it might contain carbides, is mostly matrix residue. If the authors have evidence to the contrary, they should present it in the paper but this aspect must at least be discussed.

References are missing as listed below:

l.258-262: Reference should be provided regarding the 1/3 power law for coarsening (LSW theory).
l.267: "According to the results of previous studies" References of those studies must be provided.
Table 3: References shoud be provided for the values presented in this table

Several sentences are unclear:

l.116-117: "with constant gradient" unclear, please rephrase.
l.127-129: unclear, please rephrase.
l.197-198: "the number of deformed ferrite" incomplete or incorrect.
l.257: Ostwald ripening and coarsening meaning the same in the context of precipitation, this sentence should be clarified.
l.329-331: Incomplete or incorrect sentence, please rephrase.

Language quality is an issue throughout the paper. There is a substantial number of grammar, syntax and spelling errors, a few examples of which are given below. This problem reaches a point where it gets in the way of the scientific message and the authors must address it.

l.40: "the van der Waals’ forces" should be "van der Waals' forces"
l.42: "detects" should probably be "defects"
l.43: "a great many of"
l.52-55 : Incomplete or improper sentence.
l.54: "fracture surfaces generation" should be "fracture surface generation"
l.55: "whereas"
l.70: "M-A constituent was easier to be the crack initiation sites", improper syntax and grammar.
l.84-86: Incomplete or improper sentence.
l.86: "initation" should be "initiation"
l.116: "routine"
l.132: "nickle" should probably be "nickel"

Reviewer 2 Report

Effect of tempering temperature after thermo-mechanical control process on microstructure characteristics and hydrogen-induced ductility loss in high-vanadium X80 pipeline steel

In this work, the authors performed a thorough study on the effect of tempering temperature after TMCP on the microstructure and hydrogen-induced ductility loss for X80 pipe steel. The microstructure and grain characteristics were analyzed through SEM EBSD. The mechanical properties to identify the hydrogen-induced ductility loss is studied through tensile testing and fractography. The authors found that with increasing temperature (from 450 to 650), the hydrogen-induced ductility loss improves, peaking at the T650 steel which has the optimum resistance to hydrogen-induced ductility loss due to dispersed nano-scale vanadium carbides with <20nm in size. Beyond 650, (T700), the mean size of the carbide precipitates grows larger than 20nm which has weaker hydrogen capture capability, hence, causes loss in ductility compared to the T650. The objective of the work is clear and important. All experiments were designed properly. All data were presented clearly, and the discussions were clear. I believe this paper warrants a publication in Materials. 

Minor specific comments:

  • Figure 5 a-f : it is difficult to tell a difference between thin and thick black lines unless the reader follows the text, in other words, the figure itself does not show this clearly.
  • Please check throughout the manuscript, “Fig.” should be used at the beginning of the sentence, it is only correct to use the abbreviation within the sentence. For example, line 167 should be “Figure 4 shows…”
  • Figure captions should be more explanatory, for example, Figure 3. Schematic diagram showing dimensions of the tensile specimen used in tension test with pre-charging and in air conditions.
  • Under Materials and Methods, the authors should elaborate on the specific experimental procedure used for tensile test, such as loading rate.
  • EBSD and TEM conditions used in the study should be included under the Materials and Methods section, currently the authors have a brief discussion in the “Introduction” section, ~line 102-112. Please also elaborate under the methods sections for reproducability of the readers.

Reviewer 3 Report

The authors' intention was to fill the gap regarding knowledge about influence of tempering process on the resistance to hydrogen-induced failures in X80 pipeline steel.

The article was prepared at a good technical and editorial level. An important issue of hydrogen embrittlement is addressed in it.

Comments:

  • Lines 105-112 describing material analysis methods should be moved to next section (materials and method).
  • Line 110, 114. What means: 'experimental steels' ? X80 is a well known high strength low alloy (HSLA) steel produced by thermo-mechanical control process (TMCP), widely used to transport oil and natural gas.
  • Fig.1 caption. There isn't unit for 'Time' axis.
  • Lines 178-179. I wonder if this is an observation or a guess ('would bind') ?
  • Line 182. These fine precipitates should be marked in SEM images.
  • Fig.7, Fig.10, Fig.14. Axis descriptions are illegible.
  • Units description in Charts and Tables should be unified, e.g.: stress, MPa; t, °C (instead of: ... (MPa), .../°C, ...(%), .../%).
